# Low Atrial Rhythm in a Large Cohort of Children from Transylvania, Romania

**DOI:** 10.3390/life12111895

**Published:** 2022-11-15

**Authors:** Anne-Kathrin Henckell, Gabriel Gusetu, Radu Rosu, Dana Mihaela Ciobanu, Sabina Istratoaie, Lucian Muresan, Cecilia Lazea, Dana Pop, Gabriel Cismaru, Cristian Bârsu, Stefan Lucian Popa, Alina Gabriela Negru, Andrei Cismaru, Dumitru Zdrenghea, Simona Sorana Cainap

**Affiliations:** 15th Department of Internal Medicine, Cardiology Rehabilitation, Iuliu Hatieganu University of Medicine and Pharmacy, 400012 Cluj-Napoca, Romania; 2Department of Diabetes and Nutrition Diseases, Iuliu Hatieganu University of Medicine and Pharmacy, 400012 Cluj-Napoca, Romania; 3Department of Pharmacology, Toxicology and Clinical Pharmacology, Iuliu Hatieganu University of Medicine and Pharmacy, 400012 Cluj-Napoca, Romania; 4Cardiology Department, Emile Muller Hospital, 68100 Mulhouse, France; 5Department of Pediatrics I, Emergency Clinic Hospital for Children, Iuliu Hatieganu University of Medicine and Pharmacy, 400012 Cluj-Napoca, Romania; 6Department of Abilities—Humanistic Sciences, Iuliu Hatieganu University of Medicine and Pharmacy, 400012 Cluj-Napoca, Romania; 75th Department of Internal Medicine, Medical Clinic No 2, Iuliu Hatieganu University of Medicine and Pharmacy, 400012 Cluj-Napoca, Romania; 8Department of Cardiology, Victor Babeș University of Medicine and Pharmacy of Timisoara, 300041 Timisoara, Romania; 9Research Center for Functional Genomics, Biomedicine and Translational Medicine, Iuliu Hatieganu University of Medicine and Pharmacy, 400012 Cluj-Napoca, Romania; 10Department of Pediatrics II, Emergency Clinic Hospital for Children, Iuliu Hatieganu University of Medicine and Pharmacy, 400012 Cluj-Napoca, Romania

**Keywords:** children, low atrial rhythm, right atrium, left atrium, vagal tone

## Abstract

Low atrial rhythm (LAR) is an ectopic rhythm originating in the lower part of the right or left atrium. Prior observational studies attempted to quantify the prevalence of low atrial rhythm in the pediatric population, but the observed prevalence was highly variable with relatively small sample sizes. We aimed to characterize low atrial rhythm and determine its prevalence in a large population of 24,316 asymptomatic children from northwestern Transylvania. We found a prevalence of 0.6% (145 children) for low atrial rhythm. Children with LAR had a significantly lower heart rate (mean 78.6 ± 8.3 bpm), than the control sinus rhythm group (85.02 ± 4.5 bpm). Furthermore, a shorter PR interval was seen in children with LAR (132.7 ± 12.7 ms) than in the children from the control group (141.7 ± 5.4; *p* = 0.0001).There was no significant association between gender and the presence of left LAR (LLAR) or right LAR (RLAR) (*p* = 0.5876). The heart rate of children with LLAR was significantly higher (81.7 ± 11.6 bpm) than that of the children with LRAR (77.6 ± 11.1 bpm) (*p* = 0.037). Pediatric cardiologists should recognize low atrial rhythm and be aware that asymptomatic, healthy children can exhibit this pattern, which does not require therapeutic intervention.

## 1. Introduction

Low atrial rhythm (LAR) is an ectopic atrial rhythm that originates in the lower part of the right or left atrium. The activity of the sinus node, which is the default pacemaker, is thus replaced by an ectopic atrial pacemaker. Due to the absence of symptoms, arrhythmia is diagnosed solely by ECG analysis. The retrograde P waves in the inferior leads (II, III, aVF) followed by narrow QRS complexes facilitate this diagnosis [1].

After damaging the sino-atrial node in dogs with radon, Bormax and Meek [2] were able to generate a slow atrial rhythm in the animals. They observed the origin of the cardiac impulses in the low right atrium at the level of the inferior septum close to the coronary sinus. Originally referred to as coronary sinus rhythm, this pattern was later termed low atrial rhythm. Prinzmetal et al. [3]. Stimulated the inferior part of the left atrium and obtained a low left atrial rhythm that was characterized by a negative P wave in leads II, III, and aVF. However, in healthy individuals, the rhythm is regarded as benign and typically occurs in children and young athletes with a high vagal tone [4,5]. It appears as an escape rhythm when the sinus node is too slow, or as a result of increased automaticity when it inhibits the sinus node due to its higher rate. LAR can also be associated with congenital heart defects, including left persistent superior vena cava syndrome, left atrial isomerism, and sinus venosus atrial septal Defect [6,6].

Prior observational studies attempted to quantify the prevalence of low atrial rhythm in the pediatric population. The observed prevalence was highly variable, ranging from 0% [7] to 0.21% [8], whereas some authors found a prevalence of up to 10% [9]. However, these studies had relatively small sample sizes. The optimal sample size is one that is always “representative” of the population from which it is drawn. It must be large enough to include all potential characteristics of interest and their variability within the population. Therefore, we aimed to determine the prevalence of low atrial rhythm in a large population of 24,316 asymptomatic children from northwestern Transylvania.

## 2. Materials and Methods

From May 2015 to November 2015, 24,316 ECGs were prospectively collected from students attending 12 schools in the northwestern Transylvania. The study recruited only healthy or asymptomatic children, excluding all children with known cardiac disease.

The schools were selected to accommodate, as much as possible, all age groups between six and eighteen years old, as well as a roughly equal number of girls and boys. Each ECG was recorded in the supine position at a writing speed of 25 mm/s with amplification of 0.1 mV. Each ECG recording lasted 6 s. The BTL Cardio Point CardioPoint 2.23.18524.0 (Greeneville, TN, USA) and DIAG 1.7.17214.0 ECG were utilized.

ECGs recorded by students and cardiology residents were read by pediatric cardiologists (GC, CL, CSS) and confirmed by electrophysiologists (GG, RR, LM, GC, AN). It took eight years to analyze all 24,316 ECGs (from 2015 to 2022). All subjects were subdivided into age groups according to Rinjbeek: 6–8 years; 8–12 years, 12–16 years, and 16–18 years [10].

LAR was defined by a minimum of 4 consecutive negative P waves in the inferior leads (II, III, aVF) followed by a narrow QRS complex with regular and constant intervals. Thus, a junctional AV node rhythm, with retrograde atrial depolarization, and low atrial rhythm was distinguished, based on the absence of the PR segment. Even if some authors diagnose low atrial rhythm in cases with isoelectric P waves in inferior leads, we only considered patients with inverted P waves. ECGs were then examined for concordant negative QRS complexes and T waves in leads II, III, and aVF. If this was present, it was assumed that the right arm lead was interchanged with the left leg lead. We excluded 5 ECGs from our cohort because of limb reversal. Using the analysis of R progression in the chest wall leads, all ECGs identified as LAR were also checked for dextrocardia. The ECG was classified as low left atrial rhythm (LLAR) if it met at least three of the following criteria as described by Tang et al. [11]: (1) negative P waves in DI; (2) flat or negative P wave in aVL; (3) positive P wave or “dome-and-dart” P wave in V1; and (4) negative P wave in V6 (Figure 1). The remaining ECGs were categorized as low right atrial rhythm (LRAR) (Figure 2). To avoid errors, these were randomly cross-checked and confirmed by an electrophysiologist.

### Statistical Analysis

Categorical variables, such as the presence of LAR, LLAR, RLAR, and gender were reported as counts and percentages. Continuous variables, such as age, heart rate, PR interval, and others were reported as means and standard deviation (SD) or median with the minimal and maximal values. Correlations between categorical variables such as the presence of LAR and gender were tested by a Pearson’s chi-square test. In the case of a single degree of freedom, Yate’s adjustment was performed. Categorical comparison for P wave polarity among low left and low right atrial rhythm was performed using the chi-square test. Correlations between categorical and continuous variables were carried out using a two-sample *t*-test (if a normal distribution was present), or with the Wilcoxon signed-rank test, also known as the Mann–Whitney test (which does not require any information about the distribution of the tested sample). The distribution of continuous variables was graphically presented in the form of a histogram and tested with the Shapiro–Wilk test. The choice of statistical test was made with the help of the methodological consultancy of the online data analysis program from University of Zurich. Data analysis and calculation were partly performed in Excel (Version 16.61.1) and mainly in R Studio (Version 1.0.153–© 2009–2017 R Studio, Inc.) with the language R (Version 3.1.1 2014-07-11, R.app 1.65). For the data import and export, the R package “rio” was used. The package “dyplr” helped store data in a relational database. For the graphical presentation, the package “ggplot2” as utilized. For each analysis, a *p* value <0.05 was considered statistically significant.

## 3. Results

Among the 24,316 studied subjects, 1411 (5.8%) were aged between 6 to 8 years, 8802 (36.2%) were between 8 to 12 years, 9337 (38.4%) were between 12–16 years, and 4765 (19.6%) were between 16–18 years old. Of all children, 12,596 (51.8%) were girls and 11,720 (48.2%) were boys. The youngest study participant was 6 years old, while the oldest was 18 years old (Table 1). The mean weight was 49.4 ± 15.9 kg, mean height was 155.1 ± 15.2 kg, mean BMI was 20.2 ± 4.2 kg/m2 and mean BSA = 1.39 ± 0.36 m^2^. Compared to females, males had slightly increased height, weight and BSA (*p* = 0.0039).

In the study sample, the prevalence of low atrial rhythm was 0.60% (145 children). Although the prevalence appeared nearly identical across all age categories, 5–8-year-old children had the highest prevalence of 0.85%. By dividing the age groups by gender, the lowest frequency was seen in 8–12-year-old girls (0.42%). On the contrary, the most prevalent category (1.02%) was boys aged 5–8 years (Table 2). We found no significant association between gender and the presence of LLAR or LRAR (Chi2 = 0.2941, *p* = 0.5876).

Children with LAR had a significantly lower heart rate (mean 78.6 ± 8.3 bpm), than the control sinus rhythm group (85.02 ± 4.5 bpm). Furthermore, a shorter PR interval was seen in children with LAR (mean PR interval 132.7 ± 12.7 ms) than in the children of the control group (141.7 ± 5.4; *p* = 0.0064). Additionally, a significant difference was also found in the P wave axis: while the children with LAR had a negative axis (−50.1 ± 102.5°), the children in the control group have a positive axis (43.1 ± 83.9°) (Figure 3). However, there was no difference in P wave duration between the 2 groups (Table 3).

We also compared children with LLAR, 35 children (24.1%), and children with LRAR, 110 children, (75.1%). The heart rates of children with LLAR were significantly higher (mean HR: 81.7 ± 11.6 bpm) than those of the children with LRAR (mean HR: 77.6 ± 11.1 bpm) (*p* = 0.037). The P wave axes of the children with LLAR were different (mean P axis: −88.9 ± 47.2°) from those of children with LRAR (mean P axis: −38 ± 52.3°). The PR intervals in children with LRAR were longer (mean PR interval: 136.5 ± 22 ms) than in children with LLAR (126 ± 20 ms) (*p* = 0.046) (Table 4). However, no relation could be detected between the heart rate and the P-R interval in children with LRAR or LLAR.

## 4. Discussion

Our study shows that 0.60% of the analyzed 24,316 children from northwestern Transylvania presented LAR in their resting ECG. No significant difference was observed between different age groups or genders. Mancone et al. found a similar prevalence rate of 0.5% among 11,949 students aged 13 to 19 years old. [12]. Vilardell et al. studied 1911 asymptomatic children aged 13–14 years using a resting ECG as the diagnostic tool and observed a prevalence of 0.21% of low atrial rhythm [8]. In a study by Scherf and Harris, 23,610 electrocardiograms were performed in adults over a period of 6 years [13]. LAR was observed in 31 patients, including 17 men and 14 women, with a calculated incidence of 0.13%, lower than in our study. Furthermore, we identified 145 LAR in a shorter period of time of only 7 months. In another series of 10,000 cases a LAR was identified in 15 patients with an incidence of 0.15% [14]. Since LAR is typically an asymptomatic arrhythmia, few data exist on children, with most ECGs being performed in case of chest discomfort or symptoms suggestive of heart disease.

We ruled out alternate causes for negative P waves in precordial leads. In case of limb lead reversal, ECG resembles that of LAR. Lead I as well as the inferior leads exhibit negative P waves. In the general population, 0.4–4% of obtained ECGs on average have switched limb electrodes [15]. However, these are followed by negative concordant QRS complexes and T waves, which enabled us to differentiate from LAR. Positive P waves are also visible in lead aVR [16]. We excluded 5 ECGs from our cohort because of limb reversal.

The most frequent cause of LAR is increased vagal tone. The parasympathetic nerve fibers of the 10th cranial nerve, (vagus nerve) provide a negative chronotropic and dromotropic effect on the sinus node. Thus, excessive vagal tone can downregulate the sinoatrial node to such an extent and slow the conduction of the action potential of the AV node so that an escape rhythm is generated by another pacemaker in close proximity to the AV node. This switch from the sinus node to the AV node as a pacemaker, caused by increased vagus stimulation, was replicated in an experiment by Bouman et al. [17]. However, when D’Ascenzi et al. studied 62 athletic male adolescents with ECGs, none of those studied had LAR (0%) [5]. Mancone et al. compared the prevalence of LAR in non-competitive and competitive athletes to that of the general population. The prevalence of LAR was the same among competitive athletes and the general population (0.5%), but it was even lower among non-competitive athletes (0.4%) [12]. Since children, particularly male adolescents, and endurance athletes have higher vagal tonus, LAR is expected and considered a normal ECG variant [18,19,20]. Further diagnostic testing will be conducted in athletes and children only in the case of symptomatic LAR or a suspicious family history [20]. One of the features of the LAR is its lability, which refers to how easily it can be converted to a sinus rhythm by means such as physical activity, the use of nitrates, or carotid sinus massage. One of the hallmarks of the LAR is its adaptability since it can quickly be converted to sinus rhythm by exercise, nitrates, and carotid sinus massage. Evidently, this change is possible in healthy individuals with proper sinus node function.

LAR can be produced in dogs by warming of the coronary sinus. Zahn et al. described an ectopic rhythm with deeply inverted P waves and a normal PR interval in dogs after warming of the coronary ostium area through the wall of the coronary vein [21]. After discontinuation of the warming phenomenon, the normal sinus rhythm re-occurred. This phenomenon can be explained by the presence of specific fibers that arise from the ostium of the coronary sinus and are distributed to the posterior part of the atrio-ventricular node as described by Tawara [22]. Furthermore, Kung described a muscular bundle containing ganglionic cells entering the atrio-ventricular node from the ostium of the coronary sinus [23]. This network of nerve fibers explains the effect of exercise, hypotension and carotid sinus massage on the occurrence of low atrial rhythm [24].

Eliska et al. induced low atrial rhythm by ligation of the circumflex artery and atrial branches with subsequent ischemia of the sinus node, intreratrial septum and Bachmann bundle. However, the postero-inferior part of the right atrium was preserved, with normal arterial supply. They obtained a LAR with inverted P waves in leads II, III and avF in 13 of 18 dogs, with a duration of LAR ranging from 5 min to 28 days after ligation. The heart rate also decreased from 176 to 118 bpm, and the PR interval decreased from 98 to 80 ms. The activation time from the caudal atrium (coronary sinus) to the cranial right atrium (sinus node) was 10 to 60 ms. The histological examination showed necrosis and connective tissue at the level of the sinus node with the inferior region of the atrium unaffected by ischemia [25].

LAR can also be produced by medications, cardiotoxic substances, ischemia and myocardial infarction. As all our children were asymptomatic, cytostatic such as cyclophosphamide [26] and cisplatin [27] as well as digoxin overdose can be excluded as causes of LAR. Prior surgery [28] followed by sinus node dysfunction may contribute to the development of LA, but this was not the case of our patients. Basis et al. documented hyperkalemia and scorpion bites as additional causes of LAR [29]. Furthermore, congenital heart abnormalities must be evaluated. LAR has also been linked to: left atrial isomerism, sinus venosus atrial septal defect (ASD), and persistent left superior vena cava (PLSVC) [30,31]. PLSVC is usually asymptomatic and one of the most common congenital venous anomaly occurring in 0.3 to 0.5% of the general population. Ito-Hagiwara studied 12 patients with PLSVC and found that a negative or biphasic P wave in lead III had 100% sensitivity and 81% specificity in predicting PLSVC presence [6].

LAR can be experimentally produced by stimulating the inferior part of the atrium. Harris et al. studied eleven patients undergoing left heart catheterization using the transseptal approach. Stimulation in the inferior part of the left atrium produced negative P waves in leads II, III and avF, V5 and V6, as well as positive P wave in lead V1. Furthermore, the stimulation of the postero-inferior part of the right atrium produced negative P waves in leads II, III, avF [32] but positive P waves in V6 and negative P waves in lead V1. According to this morphology, we could classify our ECGs into left lower and right lower atrial rhythms. Harris et al. also postulated the “dome and dart” morphology of the P wave in left atrial rhythm: the initial low-voltage, smooth component represents left atrial activation, and the high, sharp terminal component represents the right atrial activation. When the low atrial septum is paced, the P wave becomes negative in the inferior leads (II, III and avF) as the activation occurs in a caudo–cranial direction, and the P wave in lead V1 becomes positive as the left atrium is depolarized earlier than the right atrium [33].

The PR interval and the heart rate were lower in LAR than in the sinus rhythm. The PR interval during LAR depends not only on the position of the ectopic region of the inferior atrium but also on the conduction velocity [34] from the caudal to cranial atrium as well as from lower atrium to both ventricles. In experimental research on dogs, the PR interval during LAR was also shorter than during sinus rhythm [35]. Comparing the PR intervals of LLAR and LRAR, it is evident that LLAR has a shorter PR interval than LRAR. Nonetheless, due to the huge standard deviations, when compared between boys and girls of different age groups, it is not statistically significant, only in the 5–8-year-old boys group is the difference near to the significance level (*p* = 0.069). A shorter PR interval is also seen in patients with Wolff–Parkinson–White syndrome, but P waves are positive in this case.

When a lower atrial rhythm is observed, there are two main causes to investigate: an increased vagal tone or a structural disorder such as atrial septal defect or persistent left superior vena cava—all of which are evident on a basic echocardiogram. In any case, the rhythm does not necessitate therapeutic intervention. The workflow is detailed below (Figure 4):

Pediatric cardiologists are aware that high R waves in the right precordial leads or negative T waves in V1, V2 +/− V3 are normal patterns in children. Additionally, they should be informed that lower atrial rhythm is one of the ECG variants that occurs in 0.6% of asymptomatic, healthy children.

A major limitation of our study is the lack of follow-up. A repeat electrocardiogram for increased vagal tone and echocardiogram should have been performed during a 12-month follow-up to check for atrial septal defects and left superior vena cava persistence.

## 5. Conclusions

In our cohort of 24,316 asymptomatic children, the prevalence of low atrial rhythm was 0.6%. Children with LAR had a significantly lower heart rate than the control sinus rhythm group. A shorter PR interval was found in children with LAR than in children with sinus rhythm. A significant difference was also found in the P wave axis between LLAR and RLAR groups. There was no difference in P wave duration between the two LAR groups. Professionals working in pediatric cardiology should be aware of low atrial rhythm and the fact that asymptomatic and otherwise healthy children may have this ECG pattern.

## Figures and Tables

**Figure 1 life-12-01895-f001:**
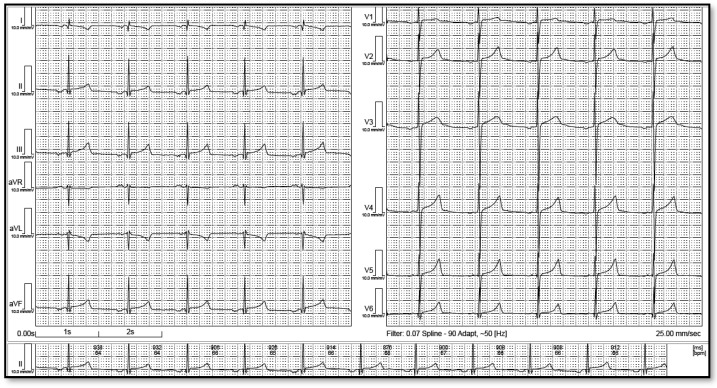
Left low atrial rhythm. Please note the negative P wave in the inferior leads as well as I and avL.

**Figure 2 life-12-01895-f002:**
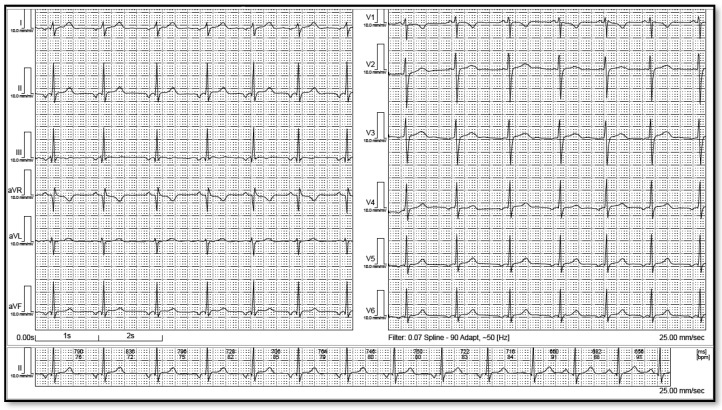
Right low atrial rhythm. Please note the negative P wave in the inferior leads and positive wave in lead I and avL.

**Figure 3 life-12-01895-f003:**
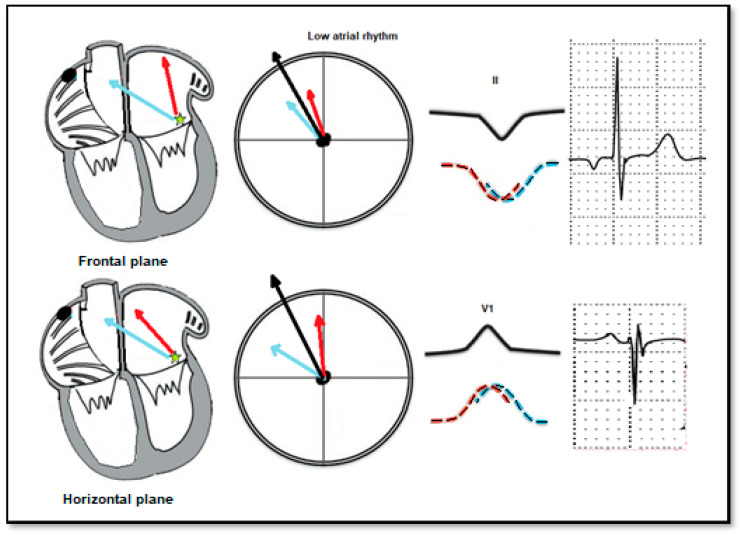
Atrial activation during left low atrial rhythm. P wave morphology and axis are different from those during sinus rhythm as the activation of both atria begins in the inferior part of the left atrium. Left atrial activation is denoted by a red arrow and right atrial activation by a blue arrow.

**Figure 4 life-12-01895-f004:**
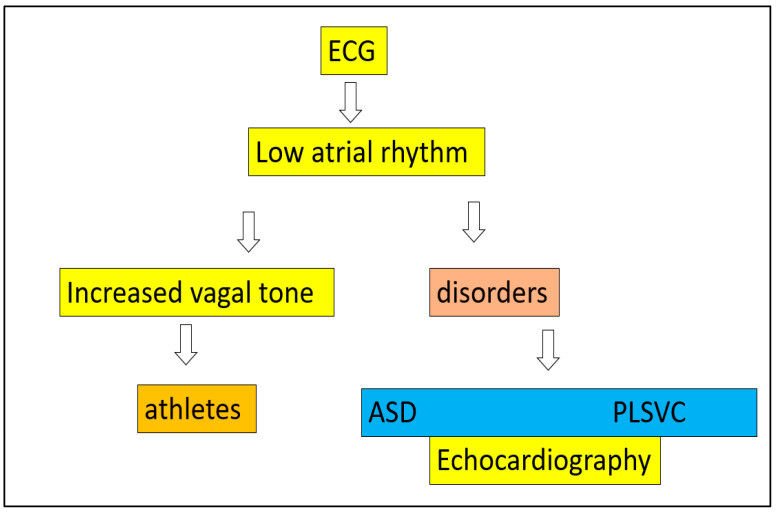
Workflow for individuals with low atrial rhythm: after an ECG reveals a low atrial rhythm, an echocardiogram is required to rule out a cardiac disorder such as ASD or PLSVC.

**Table 1 life-12-01895-t001:** Characteristics of the studied population.

	6–8 Years	8–12 Years	12–16 Years	16–18 Years
*n*	1411	8802	9337	4765
Girls No.	730	4568	4836	2463
Girls %	51.7%	51.9%	51.8%	51.7%
Mean age ± SD. median; range	7.44 ± 0.39; 7; [5.79–9]	10.09 ± 1.17; 10; [8–12]	14.06 ± 1.17; 14; [12–16]	17.16 ± 0.66; 17; [16–18]
Mean HR ± SD; median; range	92.78 ± 13.53; 92; [60–143]	89 ± 14.83; 88; [45–149]	83.5 ± 15.21; 82; [45–146]	77.44 ± 14.27; 76; [39–136]
Mean PR ±SD; median; range	133 ± 18; 132; [80–340]	138 ± 18; 136; [66–340]	140 ± 18; 136; [60–340]	150 ± 18; 148; [90–260]

**Table 2 life-12-01895-t002:** Prevalence of low atrial rhythm in the studied population.

Age Group	Girls	%	Boys	%	Total	%
5–8 years	5	0.68	7	1.02	12	0.65
8–12 years	19	0.42	25	0.60	44	0.50
12–16 years	28	0.60	31	0.69	59	0.63
16–18 years	18	0.73	12	0.52	30	0.63
Total	70	0.56	75	0.64	145	0.60

**Table 3 life-12-01895-t003:** Comparison between children with low atrial rhythm vs. control.

Of All Ages	LAR	Control	*p* Value
Mean HR ± SD (bpm)	78.6 ± 8.3	85.02 ± 4.5	0.031
Mean P duration ± SD (ms)	93.8 ± 14.7	94.2 ± 14.6	0.3264
Mean PR interval ± SD (ms)	132,7 ± 12.7	141,7 ± 5.4	0.0064
Mean P’ Axis ± SD (°)	−50.1 ± 102.5	43.1 ± 83.9	0.0022

**Table 4 life-12-01895-t004:** Comparison between children with low left vs. low right atrial rhythm.

Of All Ages	LLAR	LRAR	*p* Value
Mean HR ± SD (bpm)	81.7 ± 11.6	77.6 ± 11.1	0.03769
Mean P duration ±SD	84.3 ± 17.3	89.8 ± 16.9	0.2955
Mean PR-I ±SD	126 ± 20	136.5 ± 22	0.04654
Mean P’ Axis ± SD (°)	−88.9 ± 47.2	−38 ± 52.3	0.00344

## Data Availability

The data presented in this study are available as SPSS files on request from the corresponding author.

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
