# Peer review of "Low Atrial Rhythm in a Large Cohort of Children from Transylvania, Romania"

_life, 2022, doi:10.3390/life12111895_

Round 1

Reviewer 1 Report

This is a very very rare condition mostly asymptomatic and associated with increased vagal tone. Detail cardiological evaluation should be done to rule out any associted abnormalities.  Such children should have been followed up for at least one year. 

Author Response

Manuscript Title: “Low atrial rhythm in a large cohort of  children from Transilvania-Romania”

We would like to thank all 4 reviewers for their thoughtful review of the manuscript. They raise important issues, and their inputs are very helpful for improving the manuscript. We agree with almost all their comments, and we have revised our manuscript accordingly.

We marked with red color the modifications we have made in the revised manuscript.

Please, find below the referees’ comments repeated and our responses inserted in after each comment.

REVIEWER 1.   COMMENTS TO AUTHOR:

This is a very very rare condition mostly asymptomatic and associated with increased vagal tone. Detail cardiological evaluation should be done to rule out any associated abnormalities.  Such children should have been followed up for at least one year.

Response:  

The critic is correct. These patients should have been followed for one year. This would have been conceivable if the study had been prospective. However, our investigation was retrospective. Nevertheless, we included this aspect as Major Limitations of the study:

“The major limitation of our study is the lack of follow-up. A repeat electrocardio-gram for increased vagal tone  and echocardiogram should have been performed dur-ing a  12 months follow-up  to check for atrial septal defect and left superior vena cava persistence”

Reviewer 2 Report

Dear Sir/Madam,

I had the opportunity to act as a reviewer on the recent submission by Henckell et al. to the Life Journal.

The authors present the prevalence of low atrial rhythm in a large cohort of healthy children in Transylvania. 

The manuscript is well structured; however, some issues need to be addressed:

  1. The authors state in the abstract that “practitioners should recognize low atrial rhythm, distinguish between left and right low atrial rhythm”. First of all, what type of practitioners (i.e., GPs) and secondly, why is it important to distinguish between left and right low atrial rhythm.
  2. Please propose in the Discussion section a workflow for the work-up in children with low atrial rhythm.
  3. I recommend thoroughly reviewing the manuscript in order to define the abbreviations and to use them accordingly throughout the text. (i.e., LAR and LLAR on lines 40 and 44 not previously abbreviated; LAR on line 51 defined as the abbreviation, but not consistently used, for instance line 88).
  4. I recommend using the same nomenclature for the leads II and III throughout the manuscript (sometimes as “DII” and “DIII” described).
  5. Please review the numbering of the references (line 99 Tang et al. is found in the references at number 11 and not 12, as mentioned).
  6. Line 103: better use the term “electrophysiologists” instead of “arrhythmologists”.
  7. Table 1: please provide the percentages in boys or girls for each group and remove one of the sexes. 
  8. Please enhance the readability of Table 2, it appears very difficult to read.
  9. Lines 209-212: I recommend optimizing the text, a sentence appears to be redundant.
  10. Please provide in the Conclusion section the impact of the presented data on the daily routine.

Best regards,

Author Response

Manuscript Title: “Low atrial rhythm in a large cohort of  children from Transil-vania-Romania”

We would like to thank all 4 reviewers for their thoughtful review of the manuscript. They raise important issues, and their inputs are very helpful for improving the manuscript. We agree with almost all their comments, and we have revised our manuscript accordingly.

We marked with red color the modifications we have made in the revised manuscript.

Please, find below the referees’ comments repeated and our responses inserted in after each comment.

REVIEWER 2.   COMMENTS TO AUTHOR:

Dear Sir/Madam,

I had the opportunity to act as a reviewer on the recent submission by Henckell et al. to the Life Journal. The authors present the prevalence of low atrial rhythm in a large cohort of healthy children in Transylvania. The manuscript is well structured; however, some issues need to be addressed:

Comment No 1: The authors state in the abstract that “practitioners should recognize low atrial rhythm, distinguish between left and right low atrial rhythm”. First of all, what type of practitioners (i.e., GPs) and secondly, why is it important to distinguish between left and right low atrial rhythm.

Response: The reviewer is pretty much correct. It is unnecessary to distinguish between the right and left inferior atrial rhythms because there is no practical application for doing so. Both have no adverse pathogenic effects. To recognize that an inferior atrial rhythm exists and that it is benign is sufficient.

“Pediatric cardiologists should recognize low atrial rhythm, and be aware that asymptomatic, healthy children can exhibit this pattern that does not require therapeutic intervention”

Comment No 2: Please propose in the Discussion section a workflow for the work-up in children with low atrial rhythm.

Response: When a lower atrial rhythm is observed, there are two main causes to investigate: an increased vagal tone or a structural disorder such as atrial septal defect or persistent left superior vena cava - all of which are evident on a basic echocardiogram. In any case, the rhythm does not necessitate therapeutic intervention. A workflow is detailed below:

Figure 2. Workflow for individuals with low atrial rhythm: after an ECG reveals low atrial rhythm., an echocardiogram is required to rule out a cardiac disorder such as ASD or PLSVC

Comment No 3: I recommend thoroughly reviewing the manuscript in order to define the abbreviations and to use them accordingly throughout the text. (i.e., LAR and LLAR on lines 40 and 44 not previously abbreviated; LAR on line 51 defined as the abbreviation, but not consistently used, for instance line 88).

Response: Thank you. We corrected the abbreviations.

Comment No 4: I recommend using the same nomenclature for the leads II and III throughout the manuscript (sometimes as “DII” and “DIII” described).

Response: Thank you. We corrected the limb leads

Comment No 5: Please review the numbering of the references (line 99 Tang et al. is found in the references at number 11 and not 12, as mentioned).

Response: Thank you. We corrected the reference

Comment No 6: Line 103: better use the term “electrophysiologists” instead of “arrhythmologists”.

Response: Thank you. We corrected the term

Comment No 7: Table 1: please provide the percentages in boys or girls for each group and remove one of the sexes.

Response: Thank you. We corrected Table 1

Comment No 8: Please enhance the readability of Table 2, it appears very difficult to read.

Response: Thank you. We modified Table 2

Comment No 9: Lines 209-212: I recommend optimizing the text, a sentence appears to be redundant.

Response: Thank you. We corrected the text on the vagal tone

Comment No 10: Please provide in the Conclusion section the impact of the presented data on the daily routine.

Response: Thank you. We added the clinical impact of the findings

Reviewer 3 Report

In the manuscript of life-1924586 entitled “Low atrial rhythm in a large cohort of children from Transylvania -Romania”, Dr. Henckell and colleagues studied a prospective analysis on the P-wave form and polarity of 24,316 ECGs collected from healthy students (6 to 18 years, 51.8% of girls) to investigate the presence of ectopic rhythm originating in the lower part of the atrium, named “Low Atrial Rhythm (LAR)” by the authors. They found 145 (0.6%) students had LAR on their ECGs and they showed significantly lower heart rate (LAR versus control; 78.6 vs 85.02 bpm, P = 0.031) and shorter PR-interval (132.7 vs 141.7 ms, P = 0.0064) than control students with sinus rhythm.

Although the study has some interest, the finding was just an observation on ECGs from healthy students. The authors should describe how their findings contribute to the health checkup program or disease management. Critiques are described as following.

1.     First, the reviewer does not recognize "low atrial rhythm" as a previously authorized term. The authors referenced several papers (#8, #9, #10 and #12), but the reviewer could not find the term in these papers. If the term is presented for the first time, it should be used carefully. The reviewer is concerned that the term "low atrial rhythm" is inappropriate because it seems too short to reflect the origin of the rhythm in the lower atria.

2.     As mentioned above, describe how the findings will contribute to health screening programs and disease management. Tang et al. showed P wave configuration during atrial tachycardia can help predict the origin of arrhythmias (J Am Coll Cardiol. 1995, PMID: 7594049). Although different from such a study, it is necessary to consider and mention whether the findings of the present investigation have pathological implications. In addition, the authors noted in the simple summary as following; Our findings are significant for the society because healthcare practitioner will be reminded that an inferior atrial rhythm is rather prevalent in the pediatric population and should not be sanctioned medically (line 32-34). Why could the authors declare, without any follow-up data, that children with those ECG findings had no need for additional assessments or medications?

3.     Although the authors noted "We found no significant association between gender and the presence of LLAR or LRAR (Chi2 = 0.2941, p = 0.5876) (Table 2). (line 143-144)", the annotation of "Table 2" is inappropriate because it is not shown in the table.

4.     Like a review article, the writings in the discussion are too apart from the findings from the current investigation. Even though some speculations are permitted, it should be discussed centering the obtained results.

Author Response

Manuscript Title: “Low atrial rhythm in a large cohort of  children from Transil-vania-Romania”

We would like to thank all 4 reviewers for their thoughtful review of the manuscript. They raise important issues, and their inputs are very helpful for improving the manuscript. We agree with almost all their comments, and we have revised our manuscript accordingly.

We marked with red color the modifications we have made in the revised manuscript.

Please, find below the referees’ comments repeated and our responses inserted in after each comment.

REVIEWER 3.   COMMENTS TO AUTHOR:

In the manuscript of life-1924586 entitled “Low atrial rhythm in a large cohort of children from Transylvania -Romania”, Dr. Henckell and colleagues studied a prospective analysis on the P-wave form and polarity of 24,316 ECGs collected from healthy students (6 to 18 years, 51.8% of girls) to investigate the presence of ectopic rhythm originating in the lower part of the atrium, named “Low Atrial Rhythm (LAR)” by the authors. They found 145 (0.6%) students had LAR on their ECGs and they showed significantly lower heart rate (LAR versus control; 78.6 vs 85.02 bpm, P = 0.031) and shorter PR-interval (132.7 vs 141.7 ms, P = 0.0064) than control students with sinus rhythm.

Although the study has some interest, the finding was just an observation on ECGs from healthy students. The authors should describe how their findings contribute to the health checkup program or disease management. Critiques are described as following.

Comment No 1.     First, the reviewer does not recognize "low atrial rhythm" as a previously authorized term. The authors referenced several papers (#8, #9, #10 and #12), but the reviewer could not find the term in these papers. If the term is presented for the first time, it should be used carefully. The reviewer is concerned that the term "low atrial rhythm" is inappropriate because it seems too short to reflect the origin of the rhythm in the lower atria.

Response:  The term was earlier used  by de Pablo Marquez et al. to describe a rhythm marked by negative P waves in inferior leads II, III, and avF in a 9-year-old athlete with increased vagal tone[i]. The same authors described 2 years later a 17-year old asymptomatic patient with low atrial rhythm characterized by negative P waves in inferior leads: II, III and avF[ii]. The term was also used by Hatem Arı et al who reported the case of a 55-year old patient with negative P waves in inferior leads that mimicked acute myocardial infarction[iii]. Akyel et al. reported the case of a 49 year old male admitted for acute inferior myocardial infarction after taking propyphenazone with transitory negative P waves in inferior leads. They named the rhythm: low atrial rhythm[iv]. The information was added to the Discussion section.

Comment No 2.     As mentioned above, describe how the findings will contribute to health screening programs and disease management. Tang et al. showed P wave configuration during atrial tachycardia can help predict the origin of arrhythmias (J Am Coll Cardiol. 1995, PMID: 7594049). Although different from such a study, it is necessary to consider and mention whether the findings of the present investigation have pathological implications. In addition, the authors noted in the simple summary as following; Our findings are significant for the society because healthcare practitioner will be reminded that an inferior atrial rhythm is rather prevalent in the pediatric population and should not be sanctioned medically (line 32-34). Why could the authors declare, without any follow-up data, that children with those ECG findings had no need for additional assessments or medications?

Response:  The critic is correct. We had to remove the phrase and insert the following sentence:

“Pediatric cardiologists should recognize low atrial rhythm, and be aware that asymptomatic, healthy children can exhibit this ECG pattern”

Comment No 3.     Although the authors noted "We found no significant association between gender and the presence of LLAR or LRAR (Chi2 = 0.2941, p = 0.5876) (Table 2). (line 143-144)", the annotation of "Table 2" is inappropriate because it is not shown in the table.

Response:  Thank you. We corrected the text and added Table 2 next to age group prevalence.

Comment No 4.     Like a review article, the writings in the discussion are too apart from the findings from the current investigation. Even though some speculations are permitted, it should be discussed centering the obtained results.

Response:  Thank you. We modified the Discussion section.

[i] de Pablo Márquez B, Salvador Sáncheza J, Oliveras Vilàb T, García Lópezc C, Grange Sobea IP. Low atrial rhythm. Apunts sports medicine. 2014;48:3-4

[ii] de Pablo Márquez B, Oliveras Vilà T, Grange Sobe IP. Low atrial rhythm. Med Clin (Barc). 2016. 4;146(5):237.

[iii] Arı H, Kahraman F, Baş HA, Arslan A. Low atrial rhythm mimics myocardial infarction. Anatol J Cardiol. 2015 Aug;15(8):675, 683. doi: 10.5152/AnatolJCardiol.2015.6407. PMID: 26301351; PMCID: PMC5336872.

Bernat de Pablo Márqueza,

[iv] Ahmet Akyel ⁎, Yakup Alsancak, Çağrı Yayla, Asife Şahinarslan, Murat Özdemir. Acute inferior myocardial infarction with low atrial rhythm due to propyphenazone: Kounis syndrome. International Journal of Cardiology 2010.148:352-353.

Reviewer 4 Report

Henckell et al report the prevalence of low atrial rhythm in the pediatric population which was around 0.6% and in line with previous literature. This is a well-done prospective study describing prevalence which has been previously well established in the adult population but limited studies in the pediatric population. The diagnosis of low atrial rhythm was based on EKG and authors took steps to prevent misdiagnosis including junctional rhythm and lead reversal.  

I have the following minor suggestions. 

1.     The authors state that “The ECGs were recorded by students and cardiology residents and analyzed by pe- 85 diatric cardiologist and arrhythmologists” 

Was the ECG read by one person or confirmed by two people? I would recommend stating that for inter observer variability.

Who were the authors who read the EKG? Would mention the initials following the statement.

2.     Results - 8802 132 (36,2%) of the study subject, 9937 (38,4%) ; correct to 36.2 and 38.4 instead of comma ; this is present in multiple places, please correct

3.     Please use ± instead of +/- where appropriate; multiple places in the text where it +/-

4.     Grammatical errors;  (Mean 78.6 ± 8.3 bpm); mean in bracket should not be capitalized; please correct in all instances

Author Response

Manuscript Title: “Low atrial rhythm in a large cohort of  children from Transil-vania-Romania”

We would like to thank all 4 reviewers for their thoughtful review of the manuscript. They raise important issues, and their inputs are very helpful for improving the manuscript. We agree with almost all their comments, and we have revised our manuscript accordingly.

We marked with red color the modifications we have made in the revised manuscript.

Please, find below the referees’ comments repeated and our responses inserted in after each comment.

REVIEWER 4.   COMMENTS TO AUTHOR:

Henckell et al. report the prevalence of low atrial rhythm in the pediatric population which was around 0.6% and in line with previous literature. This is a well-done prospective study describing prevalence which has been previously well established in the adult population but limited studies in the pediatric population. The diagnosis of low atrial rhythm was based on EKG and authors took steps to prevent misdiagnosis including junctional rhythm and lead reversal. 

I have the following minor suggestions.

Comment No 1: The authors state that “The ECGs were recorded by students and cardiology residents and analyzed by pediatric cardiologist and arrhythmologists” Was the ECG read by one person or confirmed by two people? I would recommend stating that for inter observer variability. Who were the authors who read the EKG? Would mention the initials following the statement.

Response:  ECGs were recorder by students and cardiology residents, were read by pediatric cardiologists (GC, CL, CSS) and confirmed by electrophysiologists (GG, RR, LM, GC, AN). It took eight years to analyze all ECGs (2015 to 2022).

Comment No 2.     Results - 8802 132 (36,2%) of the study subject, 9937 (38,4%) ; correct to 36.2 and 38.4 instead of comma ; this is present in multiple places, please correct

Response:  Thank you. We corrected the text

Comment No 3.     Please use ± instead of +/- where appropriate; multiple places in the text where it +/-

Response:  Thank you. We corrected the text

Comment No 4.     Grammatical errors;  (Mean 78.6 ± 8.3 bpm); mean in bracket should not be capitalized; please correct in all instances

Response:  Thank you. We rectified the mistake.

Round 2

Reviewer 2 Report

Dear Sir/Madam,

Thank you for reviewing the manuscript and addressing the mentioned issues. These were adequately answered. Therefore, the manuscript seems suitable for publishing in the present form.

Best regards

Reviewer 3 Report

Thanks for the authors to respond to the previous critiques from the reviewer. All the critiques are responded clearly, and the reviewer believes that the manuscript comes up to be polished for readers.